# Strain background interacts with chromosome 7 aneuploidy to determine commensal and virulence phenotypes in *Candida albicans*

Abhishek Mishra[1,2], Norma V. Solis[3], Siobhan M. Dietz[4], Audra L. Crouch[1], Scott G. Filler[3,5], Matthew Z. Anderson[1,2,6,7]*

1 Department of Microbiology, The Ohio State University, Columbus, Ohio, United States of America, 2 Center for Genomic Science Innovation, University of Wisconsin - Madison, Madison, Wisconsin, United States of America, 3 Division of Infectious Diseases, Lundquist Institute for Biomedical Innovation at Harbor-UCLA Medical Center, Torrance, California, United States of America, 4 Cellular and Molecular Pathology, University of Wisconsin - Madison, Madison, Wisconsin, United States of America, 5 Department of Medicine, David Geffen School of Medicine, UCLA, Los Angeles, California, United States of America, 6 Department of Microbial Infection and Immunity, The Ohio State University, Columbus, Ohio, United States of America, 7 Laboratory of Genetics, University of Wisconsin - Madison, Madison, Wisconsin, United States of America

* mzanderson@wisc.edu

## Abstract

The human fungal pathobiont *Candida albicans* displays extensive genomic plasticity, including large-scale chromosomal changes such as aneuploidy. Chromosome trisomy appears frequently in natural and laboratory strains of *C. albicans*. Trisomy of specific chromosomes has been linked to large phenotypic effects, such as increased murine gut colonization by strains trisomic for chromosome 7 (Chr7). However, studies of whole-chromosome aneuploidy are generally limited to the SC5314 genome reference strain, making it unclear whether the imparted phenotypes are conserved across *C. albicans* genetic backgrounds. Here, we report the presence of a Chr7 trisomy in the "commensal-like" oral candidiasis strain, 529L, and dissect the contribution of Chr7 trisomy to colonization and virulence in 529L and SC5314. These experiments show that strain background and homolog identity (i.e., AAB vs ABB) interact with Chr7 trisomy to alter commensal and virulence phenotypes in multiple host niches. *In vitro* filamentation was consistently reduced by Chr7 trisomy in SC5314, but this result was not consistent for 529L. Oral colonization of mice was increased by the presence of a Chr7 trisomy in 529L but not SC5314; conversely, virulence during systemic infection was reduced by Chr7 trisomy in SC5314 but not 529L. Strikingly, the AAB Chr7 trisomy in the SC5314 background rendered this strain avirulent in murine systemic infection. Increased dosage of *NRG1* failed to reproduce most of the Chr7 trisomy phenotypes. Our results demonstrate that aneuploidy interacts with background genetic variation to produce complex phenotypic patterns that deviate from our current understanding in the genome reference strain.

**Data availability statement:** The genome sequences for the 529L isolates used as part of the study have been deposited in the Sequence Read Archive under accession number PRJNA1056337. The rest of the data are available in the manuscript and in the Supporting information files.

**Funding:** This work was supported by National Institutes of Health grants 1R01AI148788 (to M.Z.A.), 1R01DE026600 (to S.F.G), and an NSF CAREER Award 2046863 to M.Z.A. Additionally, A.M. was supported by a President's Postdoctoral Scholars Program Award by The Ohio State University, and S.M.D. was supported by the Cellular and Molecular Pathology T32, T32GM135119. The funders had no role in study design, data collection and analysis, decision to publish, or preparation of the manuscript.

**Competing interests:** The authors have declared that no competing interests exist.

## Author summary

*Candida albicans* is a clinically important fungus of humans that is also part of the microbiota that typically colonizes our bodies. Increased copies of chromosome 7 (Chr7) in the genome of the reference strain of *C. albicans* enhance host colonization of the gut. We identified a third copy of Chr7 in a strain regarded as a good colonizer of the oral and genital tract that causes less disease, mimicking the increased colonization of the reference strain. Here, we demonstrate that the number of copies of Chr7 alters phenotypes differently between these strains, and three copies of Chr7 does not universally produce a strain that is only a better commensal strain. Additionally, which copy of Chr7 in this diploid species exists as the third copy influences commensal and pathogen phenotypes. Therefore, trisomic changes of the same chromosome in different *C. albicans* strains have substantially different effects.

## Introduction

*Candida albicans* is among the most common fungal species of the human microbiota where it resides in the nasal cavity, oral cavity, gastrointestinal tract, urogenital tract, and on the skin. Multiple distinct lineages of *C. albicans* can simultaneously colonize a host niche [1,2], and a single genetic lineage of *C. albicans* can be recovered from multiple niches of the same individual [1]. Dissemination of a single lineage to multiple body sites can occur via overgrowth of a colonized niche as is frequently observed during expansion of the gut-resident fungal population prior to a bloodstream infection [3]. Yet, each colonized niche is distinguished by distinct environmental conditions, host cell types, and cohabitating microbes that exert selective pressures on the resident *C. albicans* population. Therefore, the fitness of a *C. albicans* lineage in the host is a function of strain genotype and the specific anatomic niche.

Adaptation of *C. albicans* occurs primarily through asexual mechanisms of evolution during mitotic growth that include point mutations, loss of heterozygosity (LOH), and aneuploidy in the diploid genome. Aneuploidy, or imbalanced changes in chromosome copy number, alters the allele frequencies and dosage of up to hundreds of genes simultaneously. Furthermore, the homolog identity of imbalanced chromosomes can distinguish phenotypic outcomes because of heterozygosity between the two homologs, which is roughly 1 position per 330 basepairs in the SC5314 genome reference strain [4]. Aneuploid chromosomes can arise transiently to increase fitness and then be subsequently lost when they are no longer advantageous. For example, trisomy for chromosome 5 (Chr5) often occurs in strains exposed to the antifungal drug fluconazole, which increases the copy number of the azole drug target, *ERG11*, and the transcriptional activator of drug efflux, *TAC1* [5–7]. However, maintenance of a Chr5 trisomy incurs a growth defect relative to diploid cells and is selected against in the absence of fluconazole [6,8]. Indeed, cells harboring single chromosome

trisomies generally suffer a fitness cost under nutrient-rich conditions compared to euploid cells [8–10], suggesting that aneuploidy is maintained only under specific adaptive conditions.

Trisomy of specific chromosomes is linked to improved fitness during *C. albicans* infection. In mice, Chr6 trisomy arises frequently during infection of the oral cavity [11], and Chr7 trisomy is linked to increased colonization in the gastrointestinal tract [12,13]. While Chr6 trisomy was assayed exclusively in the SC5314 genome reference strain, *C. albicans* strains harboring a Chr7 trisomy from three genetic backgrounds improved fitness in the murine gut compared to their disomic counterparts [13]. Consistent effects of chromosome trisomy for drug resistance and colonization suggest that *C. albicans* aneuploidy may produce similar phenotypes across strain backgrounds. In these cases, the imbalanced chromosome is relatively small and expected to minimize the fitness cost associated with retaining the imbalanced DNA [14]. In humans, *C. albicans* isolates obtained from commensal niches in otherwise healthy humans are often diploid [1,15], whereas bloodstream infections in the clinic often yield highly aneuploid strains that may be linked to antifungal drug exposure [16,17]. Focused investigation of defined aneuploid cells in mouse models have been carried out primarily in the genome reference strain and are limited to investigations of gut colonization in murine models. When strain fitness is assayed in multiple niches, a trade-off is often observed between colonization of a commensal niche and virulence in a bloodstream infection [16,18–21]. However, this relationship in niche fitness has been documented mostly in defined mutants and has not been explored as deeply in aneuploid strains. As a result, questions remain as to the benefit provided by aneuploidy *in vivo*, its dependence on strain background, and its interaction with various host niches.

The dichotomy ascribed to *in vivo* phenotypes has led to *C. albicans* strains being classified as either "commensal-like" or "virulent". For example, the genome reference strain, SC5314, is considered a virulent strain that causes significant mortality during bloodstream infections in mice. In the oral cavity, SC5314 is rapidly cleared [22], likely because it forms filaments that invade and damage the host epithelium, leading to a robust immune response [23]. Conversely, the "commensal-like" 529L strain, which was isolated from a patient with an oral infection, forms stunted and incomplete hyphae that induce a weak immune response. As a result, it is not cleared and is able to stably colonize mucosal sites of the host [22,24], but this strain has not been tested in a systemic model of disease. With a few exceptions [16,25], the genetic basis for such differences in niche colonization and virulence between strains is largely unknown. Here, we identified that 529L harbors a trisomic copy of Chr7, which has been implicated in promoting commensalism in the SC5314 genome reference strain. Focused dissection of the phenotypic role for Chr7 in 529L and SC5314 demonstrate that it plays unique *in vitro* and *in vivo* roles that are dependent on the strain background and the homolog that is present in excess. Together, this work shows that Chr7 genetically interacts with the *C. albicans* strain background to alter oral colonization and systemic disease that does not necessarily produce a trade-off in fitness between commensalism and virulence.

## Results

### 529L populations are dominated by Chr7 trisomy

Whole genome sequencing of the "commensal-like" 529L strain maintained in our lab revealed an aneuploid karyotype. Read depth indicated that Chr7 was present in three copies, whereas all other chromosomes were present in two copies. Yet, the published genome for 529L did not detect aneuploidy [26], and reanalysis of the deposited reads confirmed that this strain was a true diploid. We sequenced 529L stocks and single colonies donated by two *C. albicans* labs and found all contained Chr7 trisomies as well (20 of 20 colonies). Fortunately, stocks of the original isolation for 529L from a third lab comprised a mixed population with 9/10 colonies containing a Chr7 trisomy (Chr7x3) and a single colony that was fully diploid (Fig 1). Variant positions on Chr7 in the diploid 529L strain were heterozygous along the entire length of the chromosome, indicating one copy of each chromosome homolog was retained in the Chr7 disomic 529L strain.

Isolation of a 529L strain that is disomic or trisomic for Chr7 allowed for a focused investigation of the role of this chromosome in fitness across host niches and persistent oral colonization of mouse models specifically, which is a defining feature of 529L. To determine if phenotypes associated with Chr7 trisomy were specific to 529L or conserved in *C.*

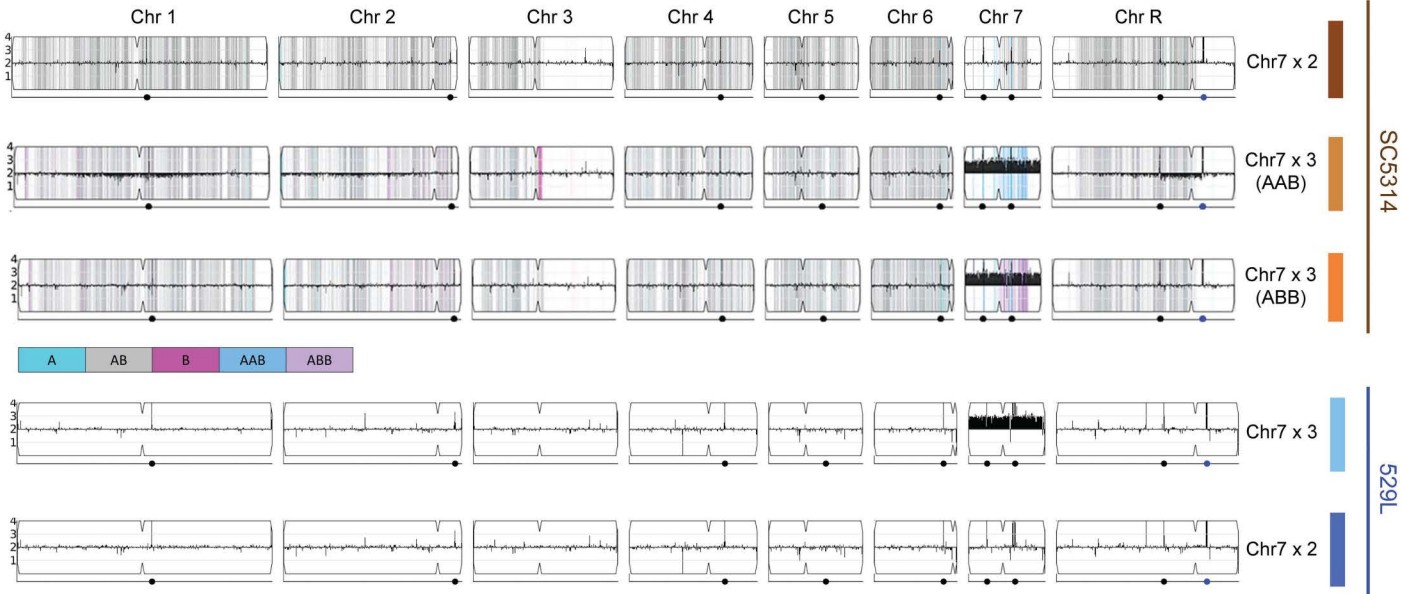

**Fig 1. Isolation of *Candida albicans* strains disomic and trisomic for chromosome 7.** Results of whole genome sequencing of five strains from two genetic backgrounds (SC5314 and 529L) were visualized using the YMAP platform [27] . Copy number is indicated on the vertical axis such that Chr7 trisomy is reflected in black bars above the lengthwise 2N midpoint of each chromosome. For SC5314 strains, homologs A and B for each chromosome based on the reference genotype are indicated by cyan and magenta, respectively. Heterozygous positions are marked in grey.

*albicans*, we obtained a diploid SC5314 strain and two aneuploid variants [8], each harboring a single trisomic copy of the A or B homolog for Chr7 (Chr7x3 AAB or Chr7x3 ABB, respectively). The Chr7 trisomy in SC5314 was confirmed using a multiplexed PCR to validate previous whole genome sequencing (WGS) of these strains (Figs 1 and S1) [8]. Together, these five strains (SC5314: Chr7x2, Chr7x3 AAB, Chr7x3 ABB and 529L: Chr7x2, Chr7x3) could be used to tease apart background-specific and homolog-specific effects of Chr7x3 on relevant *C. albicans* phenotypes.

### *In vitro* phenotypes altered by Chr7 trisomy are dependent on genetic background and homolog identity

Aneuploid cells often experience cell cycle delays and have reduced fitness under optimal growth conditions for diploid strains [28]. In *C. albicans*, aneuploid strains harboring a trisomic chromosome generally display growth defects compared to their euploid counterparts, but this has only been tested in the SC5314 strain background [8]. To test for *in vitro* growth defects in Chr7x3 strains of SC5314 and 529L, cells from each disomic and trisomic strain were cultured in liquid YPD at 30°C through exponential phase. Growth of the SC5314 Chr7x3 AAB strain differed substantially from the diploid, but the growth kinetics of the SC5314 Chr7x3 ABB strain generally mirrored the diploid parental strain (Fig 2A). There was a trend towards reduced doubling time of both Chr7x3 SC5314 strains that was not statistically different from the diploid (Fig 2B). Instead, the altered growth of the SC5314 Chr7x3 AAB strain was most evident in its reduced carrying capacity (Fig 2B). In contrast to SC5314, Chr7 trisomy of 529L did not alter either the doubling time or carrying capacity during growth in rich medium.

Strains in an SC5314-derived background (SN152) containing a trisomic copy of Chr7 are particularly sensitive to growth on medium-chain fatty acids [29]. To test the conservation of this susceptibility across strain backgrounds, each of the disomic or trisomic strains were grown across a series of undecanoic acid ($C_{11}$) or undeca-10-noic acid (UDA) concentrations, ranging from 1 µg/mL to 1 mg/mL, in YPD rich medium for two days. The optical density for liquid cultures was determined at day 1 and day 2 and was generally consistent between the two time points. Increasing concentrations

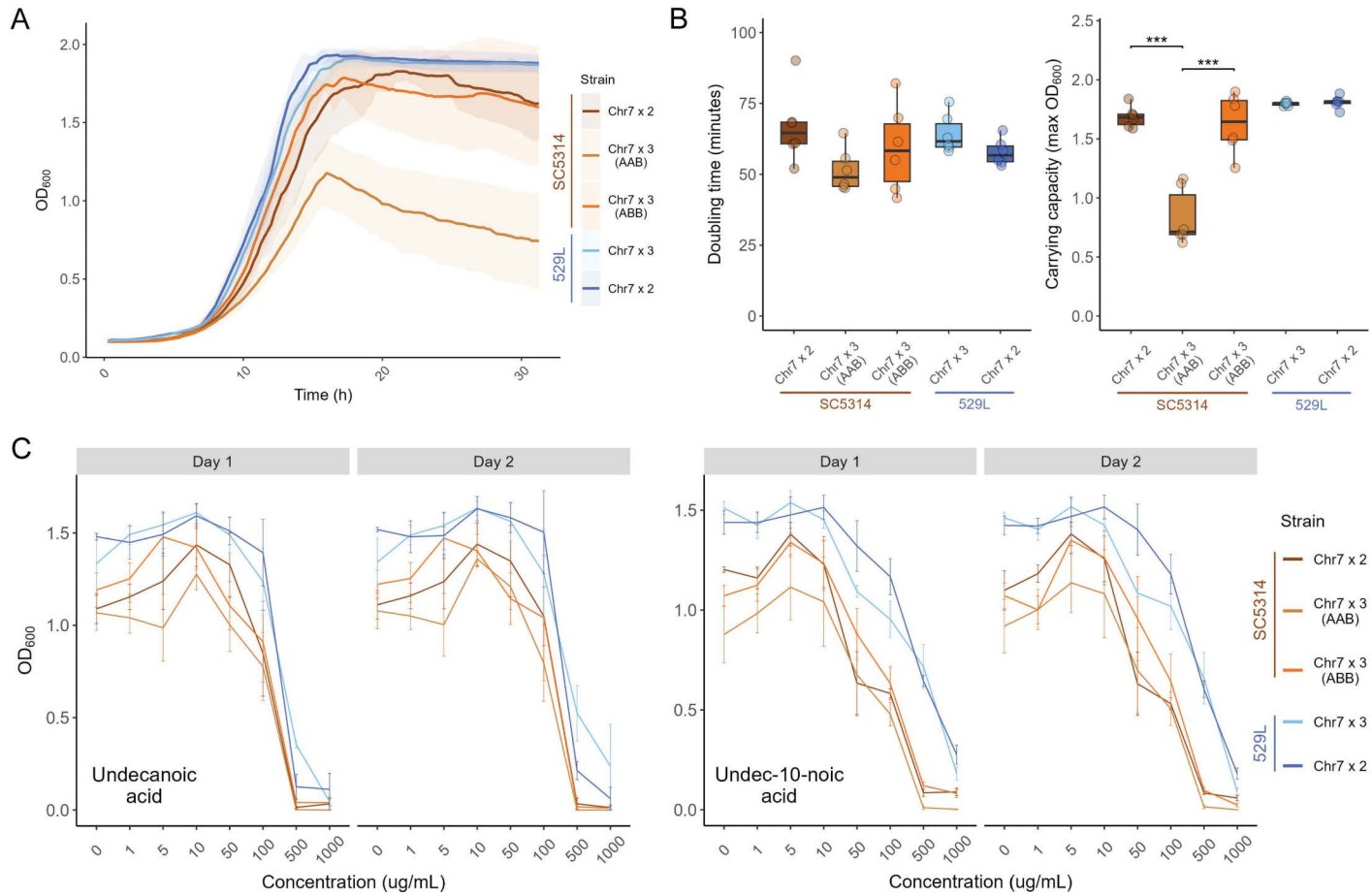

**Fig 2. Growth in rich medium and medium-chain fatty acids is minimally affected by Chr7 trisomy.** (A) Each strain was grown in liquid YPD at 30°C for 31 h. The average optical density is plotted as a thick line with the 95% confidence intervals shaded. N = 6. (B) The fastest doubling time and the carrying capacity (max OD) was determined from each growth curve and plotted. Boxplots represent the interquartile ranges with the median marked and whiskers extending to the outermost data points up to 1.5 times the interquartile range. One-way ANOVAs for each strain background, followed by Tukey's HSD test (*** represents $p < 0.001$). (C) The optical density of each strain was determined after 48 h of growth across a range of concentrations of medium-chain antifungal fatty acids, undecanoic acid and undec-10-noic acid. Means are plotted with standard error. N = 3 (two-way ANOVAs for each strain background for each day, with *strain* and *concentration* as the two fixed factors).

of either fatty acid reduced the growth of all *C. albicans* strains, and SC5314 had lower cell densities at all medium-chain fatty acid concentrations compared to 529L (Fig 2C). However, no difference in total growth existed between Chr7x2 and Chr7x3 strains from either genetic background. Therefore, trisomy of Chr7 does not alter sensitivity to these medium chain fatty acids under these conditions.

The ability to form hyphae is a primary virulence determinant in *C. albicans* that varies between genetic backgrounds [1,16,30–33]. The genome reference strain SC5314 filaments robustly under a variety of *in vitro* and *in vivo* conditions, whereas 529L typically produces stunted filaments and remains in the yeast state under hyphal-inducing conditions [30]. To determine if the presence of a trisomic Chr7 is associated with these canonical strain differences in filamentation, cells from overnight cultures were grown in liquid RPMI at 30°C for 3 hours and categorized as yeast or filamentous (*e.g.,* hyphal or pseudohyphal). SC5314 diploid cells filamented robustly in RPMI, whereas both Chr7x3 AAB and Chr7x3 ABB strains displayed significantly reduced filamentation (Fig 3A and 3B, $\chi^2 = 24.8$, p < 4.13E-6). The Chr7x3 strain of 529L also

displayed reduced filamentation responses compared to the 529L diploid ($\chi^2 = 6.9$, $p < 8.81E-3$), despite substantially lower levels of filamentation compared to SC5314.

Filamentation responses are context dependent and can differ between liquid and solid phases for the same strain [34]. Filamentation of the SC5314 and 529L strains was assessed on solid agar media to determine if the reduced filamentation associated with Chr7 trisomy in liquid medium was again observed. Approximately 100 cells were plated on either YPD or Spider agar media and incubated for 5 days at 30°C or 37°C to allow colonies to form. Filamentation was determined as the ratio of areas between radial filamentation and the central yeast colony [30]. In general, filamentation on solid YPD medium was substantially lower than on solid Spider medium for all strains (Fig 3C and 3D). The presence of a Chr7 trisomy did not alter filamentation on YPD medium for SC5314, but the Chr7x3 variant of 529L displayed reduced hyphal formation compared to the 529L diploid at 30°C (Fig 3D). On solid Spider medium, the Chr7x3 SC5314 strains filamented up to 50% less than the SC5314 diploid, but only the Chr7x3 AAB variant was statistically different from the SC5314 diploid. The Chr7x3 AAB SC5314 strain was also significantly less filamentous than its Chr7x3 ABB SC5314 counterpart. This is unlikely to be caused by a general growth defect as the colony diameter was similar across SC5314 karyotypes on solid Spider medium at 37oC (S2 Fig). Conversely, Chr7 trisomy had no effect on the relatively minor filamentation

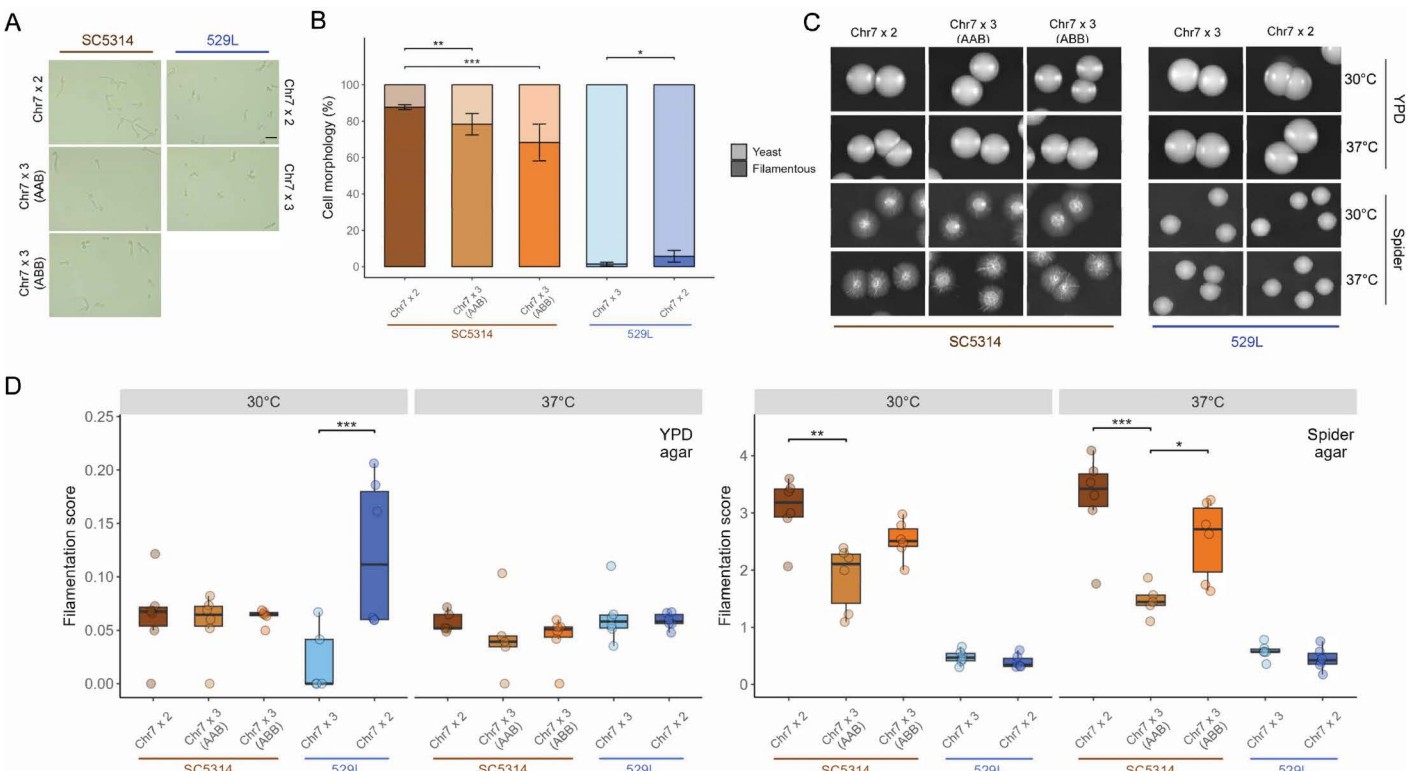

**Fig 3. *In vitro* filamentation is reduced by Chr7 trisomy in both strain backgrounds.** (A) Cellular morphology of the five strains grown under hyphal-inducing conditions, liquid RPMI at 30°C, for three hours. Scale bar = 20 μm. (B) At least 50 cells across 5+ fields of view were categorized as either yeast or filamentous for each strain per replicate and plotted as the average with standard error. Binomial GLMs for each strain background; * for $p < 0.05$, ** for $p < 0.01$, and *** for $p < 0.001$. N = 3. (C) Between 80-120 cells were plated to rich (YPD) and hyphae-inducing (Spider) solid media and allowed to grow for 5 d at 30°C and 37°C. Representative colony morphologies of the five strains are shown. (D) The filamentation score for colonies from each plate was quantified using the area of radial filamentation and center yeast colony as described in [30]. Boxplots represent the interquartile ranges with the median marked and whiskers extending to the outermost data points up to 1.5 times the interquartile range. One-way ANOVAs for each strain background; * for $p < 0.05$, ** for $p < 0.01$, and *** for $p < 0.001$. N = 6.

phenotype of 529L (Fig 3C and 3D). Each 529L karyotype was relatively stable following this assay. Sequencing of ten colonies from the Chr7 trisomic variant always revealed the extra copy of Chr7, and sequencing of three colonies from the disomic 529L strain produced only diploid genotypes. Taken together, these results indicate that Chr7 trisomy generally reduces filamentation in both liquid and solid contexts for both strain backgrounds, and that the trisomic homolog influences the degree of filamentation loss in SC5314.

## Strain differences drive oral epithelial interaction and are influenced by chromosome homolog

To test if differences in Chr7 trisomy identified between SC5314 and 529L stocks are responsible for their opposing colonization phenotypes in the murine oral cavity [22,23,35], we first attempted to recapitulate the colonization and damage phenotypes of these strains in a simplified oral epithelial cell culture model through their adherence to the epithelial cells, internalization into epithelial cells, and damage of the epithelial monolayer. Diploid SC5314 and 529L cells were inoculated onto OFK6/TERT-2 oral epithelial cell monolayers pre-loaded with $Cr^{51}$ at an MOI of 5. Strains adhered to the oral epithelium with equal efficiency at 2.5 hours post inoculation (Fig 4A and 4B), but SC5314-derived strains displayed markedly greater uptake associated with filamentation and caused substantially more damage to the epithelial layer than 529L strains after 7 hours (Fig 4A, 4C and 4D). Thus, epithelial cell culture generally modeled aspects of filamentation and damage observed for these strains during oral colonization of mice.

The ability to replicate fundamental aspects of murine oral colonization with OFK6/TERT-2 cell culture provided a platform for the investigation of Chr7 in filamentation and epithelial damage between disomic and trisomic strains. Relative to the disomic strains, Chr7 trisomy did not significantly alter adherence to host cells for either SC5314 or 529L, although the Chr7x3 ABB variant of SC5314 had significantly fewer fungal cells attached to the oral epithelium than the Chr7x3 AAB variant (Fig 4A and 4B). Interestingly, Chr7 trisomy reduced internalization of *C. albicans* cells by the epithelial monolayer in both backgrounds (Fig 4A and 4C), suggestive of improved colonization without eliciting an immune response by penetrating into or being taken up by the tissue. Furthermore, the increased adherence of SC5314 Chr7x3 AAB cells did not lead to elevated internalization relative to SC5314 Chr7x3 ABB cells. Trisomy of Chr7 had little effect on the already minimal damage to the oral epithelium by 529L, but the SC5314 Chr7x3 ABB variant damaged the epithelial monolayer less than either the diploid or Chr7x3 AAB variant (Fig 4D). This suggests a homolog-specific effect of Chr7 on epithelial damage in SC5314 that could increase colonization by eliciting less of an immune response that prompts fungal clearance. Thus, clear differences in host interactions between SC5314 and 529L can be recapitulated in culture with oral epithelial cell lines, and Chr7 trisomy contributes to altering epithelial cell phenotypes.

## Chr7 trisomy of 529L but not SC5314 increases murine oral colonization

Tissue culture models lack the diverse epithelial cell types and host immunity of the oral cavity when assessing colonization and pathogenesis by *C. albicans*. To more directly determine if Chr7 trisomy supports oral colonization of *C. albicans*, C57BL/6 male mice were orally inoculated with $2x10^7$ cells of each SC5314 and 529L strain genotype individually. At 1- and 5-days post-infection, 10 mice were sacrificed, and the tongues were removed to calculate organ weight and fungal colonization by plating tongue homogenates for colony forming units (Fig 5A). Tongue weight was unaffected by *C. albicans* strain background, Chr7 copy number, or time point during oral infection (Fig 5B). In contrast, fungal colonization of the tongue differed substantially between SC5314 and 529L. At Day 1, 529L tongue homogenates had higher colony forming units (CFUs) than SC5314. Reduced SC5314 CFUs at Day 5 increased the disparity in colonization between SC5314 and 529L and indicated clearance of SC5314 from the host (Fig 5C). Chr7 trisomy had no effect on the infection dynamics of SC5314 but promoted increased colonization of the tongue for 529L on Day 5 (Tukey's post hoc test, $p < 1.0E-03$).

The increased fitness of Chr7 trisomic strains during gut colonization is linked to reduced hyphal morphogenesis and penetration through the epithelial lining [12,24]. Invasion of the epithelial lining by *C. albicans* recruits immune cells to the site of infection, leading to the production of pro-inflammatory cytokines and eventual fungal clearance [36]. We

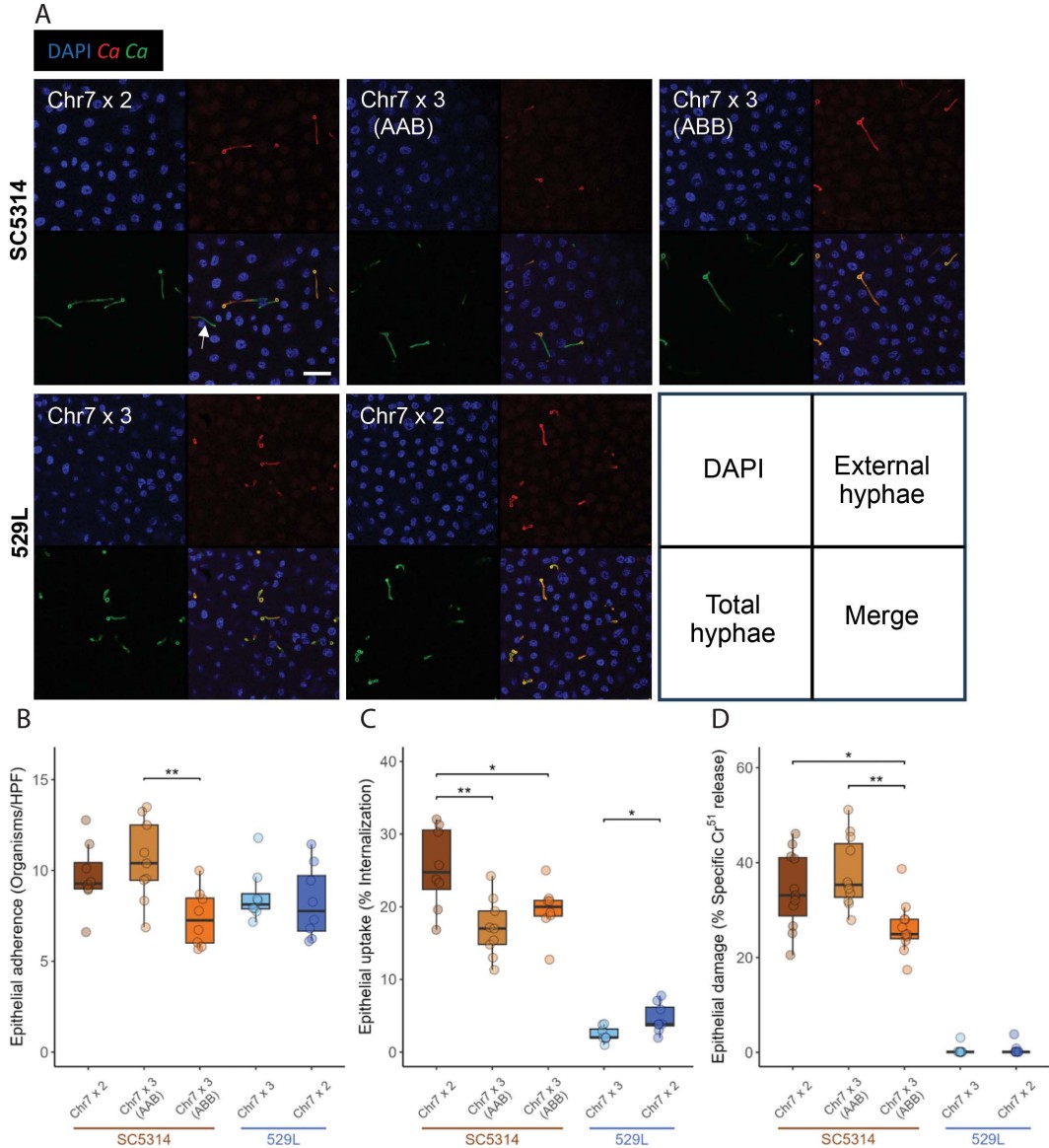

**Fig 4. Only Chr7x3 ABB trisomy in SC5314 altered epithelial cell interactions.** (A) Confocal microscopy of internalized *C. albicans* by oral epithelial cells after incubation for 2.5 h. Nuclei from all cells are visualized using DAPI, *C. albicans* (*Ca*) in green, and extracellular *C. albicans* (*Ca*) in red. An example of an internalized *C. albicans* cell segment is marked with an arrow. Scale bar 50 μm. (B) *C. albicans* yeast from each strain were incubated with OKF6/TERT-2 cells for 2.5 h. Monolayers were washed to remove non-adherent cells and the *C. albicans* remaining were determined per field of view. HPF = High Power Field. N = 9. (C) Infected monolayers were labeled by rabbit anti-*C. albicans* antiserum conjugated with Alexa 488, permeabilized, and stained with mouse anti-*C. albicans* antiserum conjugated to Alexa 568 to determine internalization. 100 cells were assayed across 10-12 fields of view. N = 9. (D) OKF6/TERT-2 cells pre-loaded with $Cr^{51}$ were incubated with $2x10^5$ *C. albicans* cells from each strain and allowed to interact. After 7 h, the supernatant of each culture was collected and the $Cr^{51}$ released determined by autoradiography. All boxplots represent the interquartile ranges with the median marked and whiskers extending to the outermost data points up to 1.5 times the interquartile range. One-way ANOVAs within the two strain backgrounds for each assay, followed by Tukey's HSD test (* for $p < 0.05$, ** for $p < 0.01$). N = 9.

hypothesized that SC5314 infection would produce high levels of inflammation, and 529L would elicit a significantly weaker inflammatory signature based on their different filamentation responses and fungal burdens. Indeed, 529L infected mice had lower cytokine levels during oral infection compared to SC5314 (Figs 5D and S3).

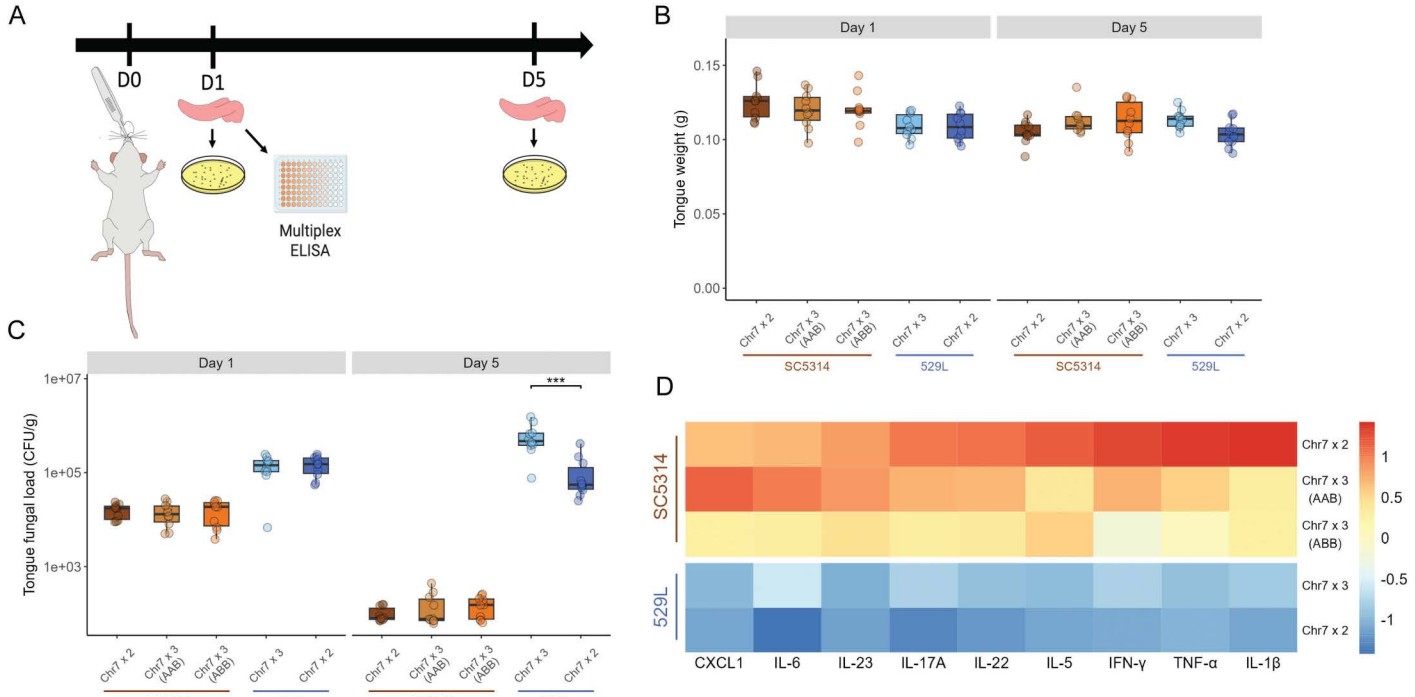

**Fig 5. Oral colonization and cytokine responses were altered by Chr7 trisomy.** (A) Oral infections of male C57BL/6 mice were initiated using cotton balls soaked with 2x10⁷ *C. albicans* cells from each strain. Tongues were collected on Days 1 and 5 post-infection from independent sets of mice and used to measure the cytokine levels or *C. albicans* abundance as indicated. (B) Extracted tongue weights of mice infected with each strain at Day 1 and Day 5. Boxplots represent the interquartile ranges with the median marked and whiskers extending to the outermost data points up to 1.5 times the interquartile range. N = 10. (C) Tongues were homogenized and 70 µL of homogenate was plated for colony forming units on YPD. Colonies were counted after 2 d growth at 30°C. Boxplots represent the interquartile ranges with the median marked and whiskers extending to the outermost data points up to 1.5 times the interquartile range. Two-way ANOVAs within each strain background with *strain* and *day* as the fixed factors, followed by Tukey's HSD test (*** represents $p < 0.001$). N = 10. (D) The abundance of eight cytokines and the CXCL1 chemokine was determined from tongue homogenates collected on Day 1. Scores were normalized within each cytokine by z-score and the average plotted on a colorimetric scale. Cytokines and CXCL1 are arranged in the order of increasing concentrations for the diploid SC5314 strain.

Interestingly, cytokine levels were also affected by Chr7 copy number despite similarity in fungal burdens within each genetic background. IL-1β, IL-22, and TNF-α levels were significantly lower in mice infected with Chr7x3 variants of SC5314 compared to Chr7x2 variants (Figs 5D and S3). Furthermore, a general trend could be observed for most cytokines across the SC5314 strains. Cytokine levels decreased for 6 of 9 assayed cytokines or the chemokine in the Chr7x3 AAB variant (IL-17a, IL-22, IL-5, IFN-γ, TNF-α, and IL-1β) and further decreased for all cytokines and the CXCL1 chemokine in the Chr7x3 ABB variant when compared to infection with Chr7x2 SC5314 (S3 Fig). Unexpectedly, Chr7x3 529L infection produced higher or equivalent levels of each cytokine or chemokine on average compared to diploid 529L although no cytokine reached statistical significance (Figs 5D and S3). It is possible that the increased cytokine levels reflect the higher fungal loads associated with Chr7 trisomy in 529L.

## Chr7 trisomy reduces SC5314 virulence but has no effect on 529L

A tradeoff in fitness between colonization of mucosal sites and virulence in systemic disease is often observed for either *C. albicans* lineages or defined mutants [16,18–20]. This dichotomy suggests that 529L would be a poor pathogen owing to its strong colonization phenotype in the gut and the oral cavity [22,24]. To investigate the dynamics of bloodstream infections by diploid SC5314 and 529L as well as the impact of Chr7 trisomy on these phenotypes, 5x10⁵ yeast of each

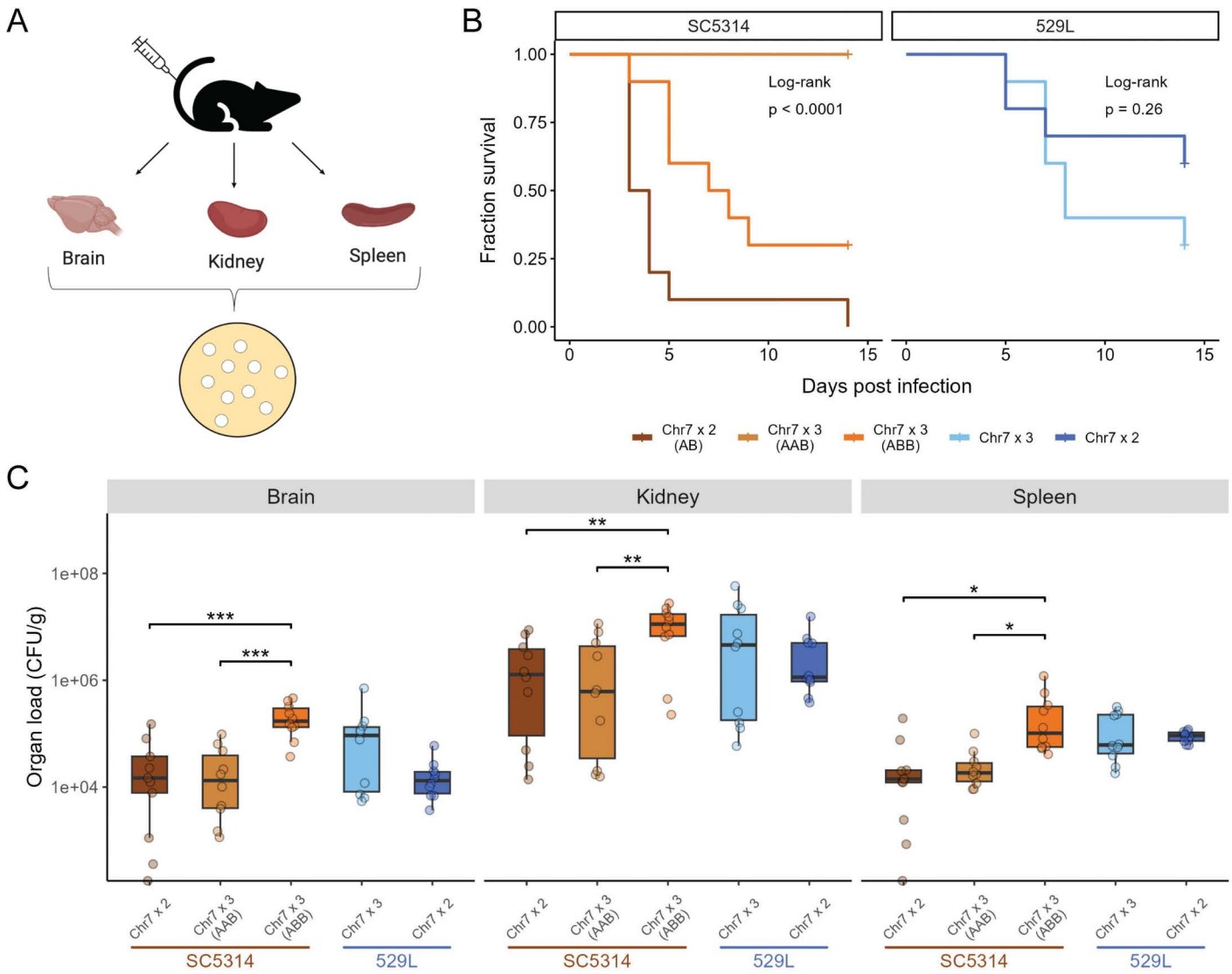

**Fig 6. Chr7 trisomy dramatically reduced systemic virulence and promoted organ colonization in SC5314 but not in 529L.** (A) C57BL/6 mice were infected with the five *C. albicans* strains via tail vein injection. One set of mice was monitored for mortality up to two weeks, while the other set of mice was sacrificed 2 days post-injection to recover their organs and obtain CFU counts. (B) Ten C57BL/6 mice (5 male, 5 female) were injected with $5 \times 10^5$ *C. albicans* cells for each of the five Chr7 disomic or trisomic strains. Survival of the mice is indicated in the Kaplan-Meier curve and significance of pairwise comparisons was determined by Log-Rank test followed by Benjamini-Hochberg correction (p < 0.01 for all three pairwise comparisons in SC5314 background). (C) The brain, kidney, and spleen were harvested from ten C57BL/6 mice (5 male, 5 female) from each strain at 2 d post-infection and organ homogenates plated to determine colony forming units (CFUs) normalized to organ weight. One-way ANOVAs for each organ within a given strain background, followed by Tukey's HSD test (* for $p < 0.05$, ** for $p < 0.01$, ** for $p < 0.01$). N = 10.

strain were injected into the tail vein in each of ten mice (5 male, 5 female, Fig 6A). As expected, mice quickly succumbed to systemic infection with diploid SC5314 (Fig 6B). Surprisingly, infection with diploid 529L also resulted in the death of four mice by day 15. A separate set of ten mice were infected in the same manner and were sacrificed on day 2 when no mortality had occurred to determine fungal loads in infected organs. Relatively similar numbers of *C. albicans* were present in the brain and kidney of SC5314 and 529L infected mice, but mice infected with 529L had higher infection of the

spleen compared to SC5314 (Fig 6C; $F_{1,18}$=7.3, $p$=0.015). Thus, 529L is still able to achieve substantial fungal burden during systemic infection despite being a highly successful colonizer of mucosal niches.

Trisomy for Chr7 altered the kinetics of virulence for SC5314 but not 529L. Chr7x3 SC5314 strains had significantly reduced virulence compared to the diploid variant (Fig 6B). Surprisingly, the Chr7x3 AAB variant of SC5314 failed to cause lethal disease in any infected mouse, and the Chr7x3 ABB SC5314 strain had an intermediate level of virulence. Despite showing an intermediate virulence phenotype, the Chr7x3 ABB variant of SC5314 achieved a significantly higher fungal burden in all three organs relative to either the diploid or the Chr7x3 AAB variant (Fig 6C). Therefore, trisomy of Chr7 increased "commensal-like" phenotypes but was dependent on the strain background and niche: increased colonization of the oral niche by 529L and decreased virulence of SC5314.

### Addition of *NRG1* does not phenocopy Chr7 trisomy in either strain background

Multiple phenotypes associated with Chr7 trisomy in SC5314 strains could be attributed to dosage of *NRG1*, a negative regulator of filamentation in C. albicans [12,37]. To examine whether *NRG1* dosage could replicate strain-specific phenotypic effects of Chr7 trisomy, we constructed diploid SC5314 and 529L strains containing a third copy of *NRG1* and assayed three phenotypes altered by Chr7 trisomy.

Trisomy of Chr7, which encodes *NRG1*, reduced filamentation *in vitro* for both strain backgrounds, although to a lesser extent for the SC5314 Chr7x3 AAB strain (Figs 3B and 7A). In the SC5314 background, the *NRG1*x3 strain filamented at lower but not significantly reduced levels compared to the diploid strain. When compared to the Chr7x3 strains, the *NRG1*x3 strain filamented at higher levels and was significantly higher compared to the Chr7x3 ABB strain (Fig 7A). Of note, *NRG1* does not contain any SNVs between the two Chr7 homologs, suggesting that homolog-specific phenotypic effects of Chr7 trisomy cannot be explained by NRG1 dosage. In 529L, filamentation was low across all assayed genotypes, and *NRG1* had no effect on filamentation. Of note, the reduced filamentation of Chr7x3 in 529L was not statistically supported in this assay and is inconsistently altered in 529L.

Addition of a third copy of *NRG1* to diploid SC5314 and 529L cells did not alter adherence or damage to epithelial cells (Fig 7B). Interestingly, *NRG1*x3 in diploid strains had opposing phenotypes in the SC5314 and 529L backgrounds. An extra copy of *NRG1* in SC5314 Chr7x2 cells recapitulated the reduced uptake by epithelial cells of the Chr7 trisomy; however, *NRG1*x3 in the 529L background increased uptake by epithelial cells relative to the Chr7x2 parental strain (Figs 7B and 4C). This increased uptake of *NRG1*x3 cells is also the opposite phenotype observed in Chr7x3 529L cells. This demonstrates that *NRG1* dosage can produce opposing phenotypes depending on the strain background but does not recapitulate most Chr7x3 phenotypes.

Finally, addition of *NRG1* in Chr7x2 strains did not phenocopy systemic infection phenotypes for Chr7 trisomic strains. While diploid SC5314 strains carrying a third copy of *NRG1* had reduced systemic virulence compared to the diploid, it failed to recapitulate either the striking loss of virulence of the SC5314 Chr7x3 AAB strain or the intermediate loss of virulence in the Chr7x3 ABB strain (Figs 6B and 7C). In the 529L background, *NRG1*x3 strains caused significantly higher mortality in mice than the diploid strain (Fig 7C), exacerbating the slightly elevated virulence of the Chr7x3 genotype (Fig 6C). These results for *NRG1*, the most likely candidate to regulate filamentation and virulence phenotypes on Chr7, suggest that dosage of additional genes or alleles contributes to these aneuploid phenotypes.

### Chr7 trisomy drives important phenotypes but is less influential than strain background

To identify the phenotypes that most clearly distinguish euploid and Chr7x3 strains, the phenotype score for each trisomic strain was normalized to their diploid counterpart and compared across assays. The distribution of phenotypes in SC5314 Chr7x3 strains was more dispersed when harboring a third copy of homolog A versus homolog B, which clustered closer to the scaled diploid (Fig 8A). The largest phenotypic change in Chr7 trisomic SC5314 strains was the increased survival during systemic infection. Decreases in filamentation and carrying capacity in YPD medium at 30°C by the Chr7x3 AAB

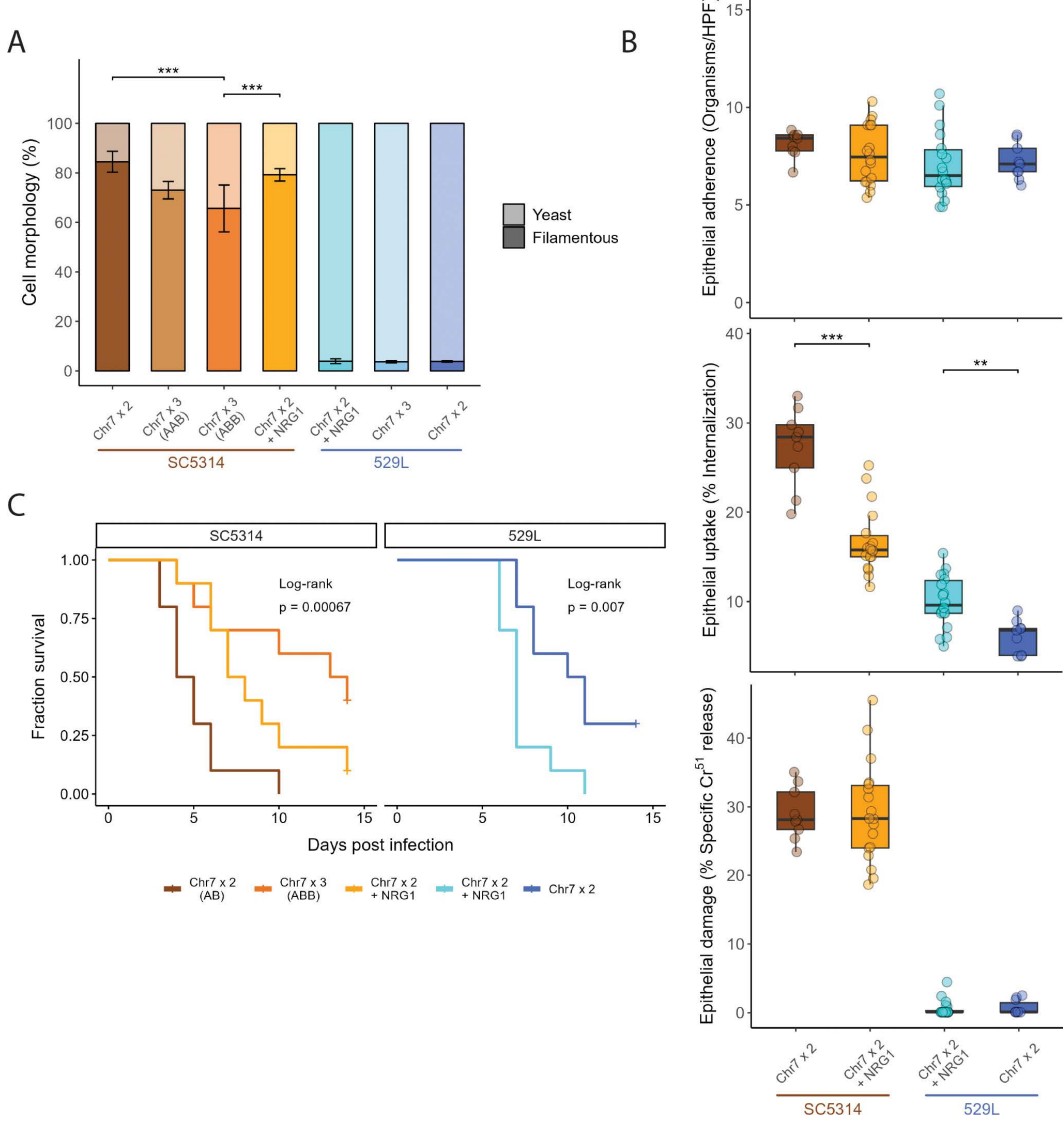

**Fig 7. Addition of *NRG1* does not phenocopy Chr7 trisomy.** (A) Diploid strains with an extra copy of *NRG1* were assayed for filamentation under hyphal-inducing conditions, liquid RPMI at 30°C, for three hours. At least 50 cells across 5+ fields of view were categorized as either yeast or filamentous for each strain per replicate and plotted as the average with standard error. Binomial GLMs for each strain background; *** for $p < 0.001$. N = 3. (B) Fungal cell adherence and uptake after co-incubation with OKF6/TERT-2 cells for 2.5 h and epithelial cell damage after 7 h were assayed. All boxplots represent the interquartile ranges with the median marked and whiskers extending to the outermost data points up to 1.5 times the interquartile range. One-way ANOVAs within the two strain backgrounds for each assay (** for $p < 0.01$, *** for $p < 0.001$). HPF = High Power Field. N = 9. (C) Ten C57BL/6 mice (5 male, 5 female) were injected via the tail vein with $5 \times 10^5$ *C. albicans* cells. Survival of the mice is indicated in the Kaplan-Meier curve and significance of pairwise comparisons was determined by Log-Rank test followed by Benjamini-Hochberg correction (p < 0.05 for Chr7x2 v. Chr7x3 (ABB) or Chr7x2 + NRG1).

SC5314 strain also exceeded the range of Chr7x3 ABB strain phenotypes, highlighting the role of dosage by specific homologs in these phenotypes. For 529L, the phenotypic range for the Chr7x3 strain was much broader than for SC5314. Increased cytokine abundance during *in vivo* oral infection and decreases in *in vitro* filamentation drove this increased phenotypic spread. Thus, the degree of phenotypic impact of Chr7 trisomy is collectively dependent on both strain background and homolog.

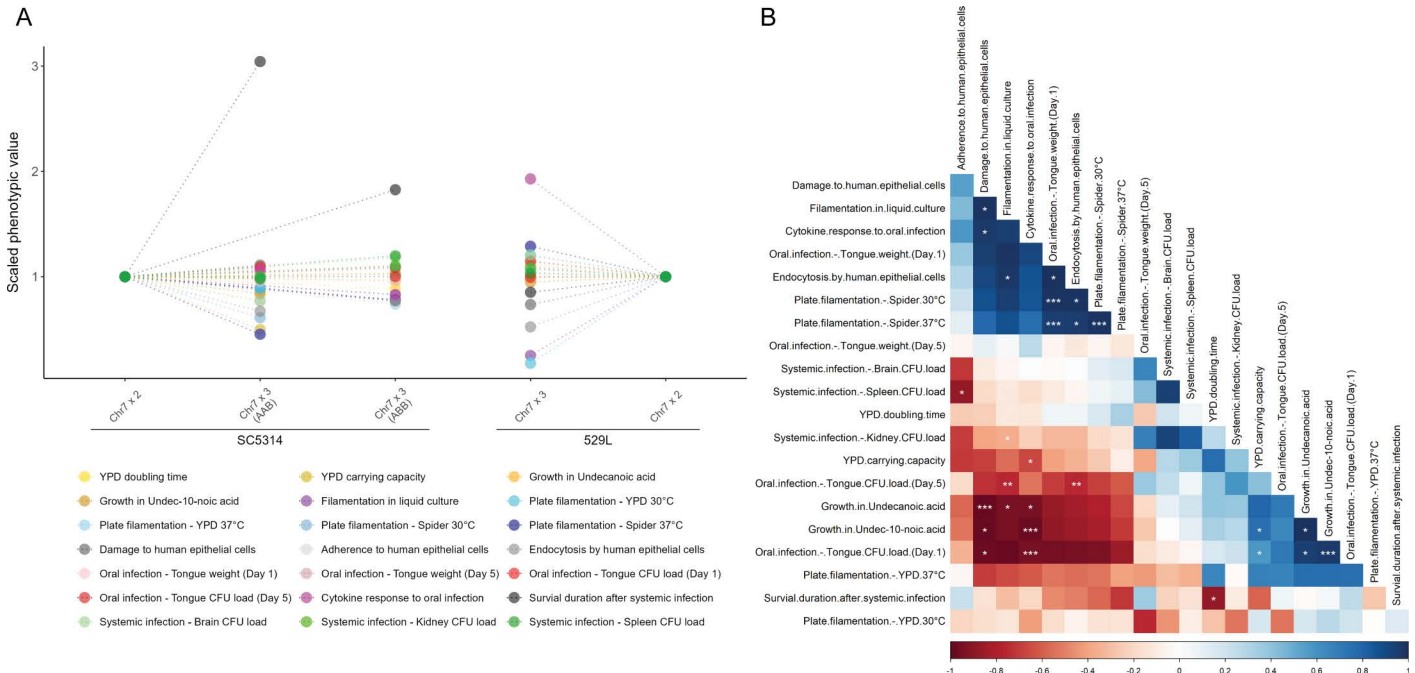

**Fig 8. Phenotypic changes due to Chr7 trisomy reveal strain-specific effects and correlated changes.** (A) Scaled phenotypic values of the Chr7 trisomic strains against the corresponding diploid from the same genetic background. The phenotypes of the two diploids were set to 1 and the aneuploid phenotypic values were linearly scaled against it, except for the CFU count assays, which were scaled on a log scale. Phenotypes are color-coded as indicated. (B) Spearman correlations were conducted between the measured phenotypes, based on all five strains used in this study. The shade of blue or red denotes the strength of positive of negative correlation coefficients, respectively, and the statistical significance is denoted with asterisks (* for $p < 0.05$, ** for $p < 0.01$, and *** for $p < 0.001$). The correlation matrix is ordered using the angular *order* of the eigenvectors (*corrplot* package in R).

Pairwise correlations between all assayed phenotypes can reveal less apparent associations between cellular responses. Correlations across all phenotypes resolved three associative clusters. The first cluster indicates a positive relationship between *in vitro* filamentation responses and oral epithelial cell damage and internalization (Fig 8B), much of which was driven by the difference between SC5314 and 529L. The second positively correlated phenotypic cluster defined differences for *in vitro* growth and oral fungal burdens between SC5314 and 529L strains one day post-infection. Both of the positively correlated phenotypic blocks are inversely associated with cytokine responses and epithelial damage phenotypes. An inverse correlation between damage and cytokine response in the oral cavity and CFU loads is driven primarily by differences between SC5314 and 529L. Of note, there was no relationship between oral colonization models *in vivo* or *in vitro* and systemic disease, suggesting these phenotypes may be separable in genetically diverse *C. albicans* isolates.

## Discussion

Commensalism and pathogenesis are complex phenotypes that arise through the contributions of multiple biological processes and are under the genetic control of many loci. Consistent mutant phenotypes in multiple strain backgrounds provide compelling evidence for a conserved biological and molecular function and avoid concerns of strain-specific effects that may not be representative of the species. While this approach has been applied to construction of defined mutants, these same principles are rarely applied to large-scale genetic changes due, in part, to the difficulty in reproducing the exact same mutation. Interrogation of commensal-pathogen phenotypes previously linked to Chr7 trisomy here demonstrate that the identity of the genetic background and the trisomic homolog are critical to interpretations of

genotype-phenotype relationships. Importantly, this principle is applicable to both relatively simple phenotypes (*e.g.,* growth) and those that are more complex, such as virulence during *in vivo* systemic disease. Furthermore, the phenotypic consequences of aneuploidy are particularly important to consider because of the frequent and spontaneous copy number fluctuation that can occur within the *C. albicans* genome, underscored by the initial observation for 529L that initiated this investigation.

A major finding from this study is that chromosomes with altered copy number do not necessarily produce the same phenotypes across genetic backgrounds. The SC5314 genome reference strain has served as the basis for nearly all studies of aneuploidy and the associated phenotypes. In some cases, an aneuploid karyotype produces similar results across *C. albicans* backgrounds (*e.g.,* Chr5 trisomy and fluconazole resistance or Chr5 monosomy and growth on sorbose) [5,38]. However, it has also been shown that an aneuploid phenotype can be restricted to a single strain (*e.g.,* Chr4 trisomy in P60002 and fluconazole resistance) [39]. Similarly, trisomy of Chr7 in SC5314 had little impact on oral colonization of mice, whereas three copies of Chr7 increased oral colonization by 529L. Therefore, assignment of aneuploid chromosomes to a given phenotype requires additional context and the underlying loci involved in producing a change may be similarly restricted to the alleles present in the aneuploid background. Furthermore, trisomy of Chr7 in 529L had no significant impact on growth in rich medium unlike trisomic strains in SC5314 that have reduced growth rates [8], which suggests the *C. albicans* genetic background may interact with phenotypes broadly ascribed to aneuploidy.

Dosage of Chr7 alters conserved and strain-specific commensal and virulence phenotypes in *C. albicans*. Experimental evolution of *C. albicans* SC5314 first pointed to a fitness advantage for strains containing a Chr7 trisomy in gut colonization and was supported by testing Chr7x3 strains from other genetic backgrounds [13]. Accordingly, Chr7 trisomic strains generally displayed reduced filamentation *in vitro*, but the slight reduction in filamentation was not consistently observed in Chr7x3 529L. Furthermore, reduced filamentation between strains did not uniformly translate to increased colonization of the oral cavity or decreased virulence during systemic infection. Instead, oral colonization was enhanced by Chr7 trisomy only for 529L, and systemic virulence was only reduced by Chr7 trisomy for SC5314. Interestingly, Chr7x3 529L strains may also have increased lethality to mice during systemic infection (Fig 6B). These conflicting phenotypes suggest that a broad characterization of Chr7 in improving colonization or commensalism is inaccurate and that both host niche and strain background are involved in determining colonization outcomes. Similarly, cytokine abundance during oral colonization was consistently altered by Chr7 trisomy but with opposing effects in the two strain backgrounds. Although the basis for reduced cytokine levels in Chr7 trisomic SC5314 strains is unclear, the increased cytokine abundances for Chr7x3 529L strains could be associated with the increased fungal burden during oral infection. While we are unable to reproduce the specific association between Chr7 trisomy and sensitivity to medium chain fatty acids, this could be a result of different contexts of exposure. Previous work used spot dilution of *C. albicans* on solid media containing the fatty acids to assay sensitivity [29], and the use of liquid medium in dilutions could alter sensitivity similar to the importance of liquid and solid contexts on filamentation responses [34]. We hypothesize that the observed phenotypic changes associated with Chr7 copy number are likely due to altered gene expression that correlates with dosage [40].

The homolog present in multiple copies for the trisomic strain plays a critical role in phenotypic consequences of aneuploidy. Phasing of the *C. albicans* SC5314 genome revealed the allele-specific expression for 147 genes [41]. When considered in combination with the fairly high frequency of heterozygous variants in *C. albicans* genomes [4,16,42], it seems likely that many phenotypes will be dependent on the alleles present in the trisomic homolog and not just aneuploidy itself. Yet, descriptions of homolog-specific phenotypes exist in only a few instances [20,43,44], and have only recently been demonstrated for trisomic chromosomes beyond drug resistance [8]. Our *in vivo* characterization of SC5314 strains that were trisomic for either Chr7 homolog demonstrated multiple phenotypes associated with an additional copy of the A or B homolog but not both. No clear pattern emerged between virulence or commensalism and a specific Chr7 homolog. For example, decreased damage to oral epithelial cells by the SC5314 Chr7x3 ABB strain does not lead to improved colonization in the oral cavity of mice. The loss of systemic virulence by the Chr7x3 AAB SC5314 strain is particularly striking,

especially given the moderately lethal phenotype of its Chr7x3 ABB SC5314 counterpart. What alleles may contribute to the loss of virulence are unclear, but these results suggest that a negative regulator of pathogenesis is likely present on this chromosome that differs between SC5314 homologs, although we must note that a small LOH on the right arm of Chr3 in Chr7x3 AAB may also contribute to the observed phenotypes.

Recent work highlighted that the reduced filamentation and increased fitness of SC5314 Chr7x3 strain in the gut was linked to dosage of the *NRG1* transcription factor [12]. Here, we also found that filamentation was lower but not significantly reduced by *NRG1* copy number in the SC5314 background, generally supporting a link between *NRG1* dosage and filamentation in this strain but that was not seen for 529L. The partially reduced virulence of *NRG1*x3 SC5314 also potentially supports a previously described role in promoting commensalism [12], but this association is inverted in 529L. Therefore, the context of *NRG1* dosage has a major impact on the phenotypic outcome. For multiple other assays not tested in [12], diploid strains in the SC5314 and 529L backgrounds engineered to carry a third copy of *NRG1* did not phenocopy their corresponding Chr7x3 variants. For example, a third copy of *NRG1* had no impact on some phenotypes altered in Chr7x3 strains (*e.g.,* epithelial adherence, epithelial damage) or did not fully mimic the Chr7 trisomic strains (*e.g., in vitro* filamentation, systemic virulence). In the case of epithelial uptake, *NRG1*x3 strains in the diploid 529L background increased epithelial uptake – the opposite phenotype of 529L Ch7x3 strains. This strongly argues that many phenotypes of Chr7 trisomic strains are the product of multiple genes and alleles that are not known to alter these processes and not just *NRG1*.

Our understand of the connection between various virulence factors and virulence in the host may be limited. Prior work in *C. albicans* suggested an inverse relationship exists between mucosal colonization and systemic disease [16,18]. In the oral cavity, successful colonization and minimal immunopathology by 529L contrasts with a strong inflammatory response and clearance of SC5314 [22] and this study. Consequently, 529L would be predicted to produce minimal virulence in a bloodstream infection. However, 529L infected tissues equally to or more robustly than SC5314 and was able to produce lethal infections. Thus, the dichotomy between commensalism and virulence is unlikely to be as clean as predicted. Instead, it may be more accurate to classify SC5314 as having colonization defects relative to other genetic backgrounds. Use of mutants or the predominance of experimentation in the SC5314 background that is uniquely filamentous may limit the robust infection of multiple host niches [43]. Furthermore, virulence of 529L and a lack of association with Chr7 trisomy suggests that *in vitro* models of colonization and damage may incompletely describe outcomes *in vivo*. A lack of filamentation or damage to tissue culture systems may not preclude the ability to transit tissues and produce fatal disease *in vivo* [45]. A greater emphasis on assessing both colonization of commensal sites and tissue burden and survival during systemic disease in the same study is needed to more clearly describe the relationship between these phenotypes in *C. albicans*. Additionally, attempts to predict virulence based on specific processes may be incomplete since they were built overwhelmingly by study in a single strain background.

Aneuploidy as a mechanism of adaptation during *in vivo* infection by *C. albicans* needs to be strongly considered because of its ability to shift disease outcomes during infection. This study adds to the growing body of evidence that certain aneuploid forms of *C. albicans* are more fit in various *in vivo* niches. Therefore, recovery of aneuploid isolates from models of infection and clinical infections may be a consequence of selection for specific karyotypes in the host and not antifungal drug exposure alone. Additionally, significant care needs to be taken in knowing the karyotype of lineages used in the lab to avoid misinterpretations of experimental results that may be confounded by aneuploidy. While the transient nature of aneuploidy imposes challenges to tracking chromosomal imbalance *in vivo*, this study highlights the importance of aneuploidy to alter *C. albicans* phenotypes relevant to commensalism and pathogenesis among strains.

## Methods

### Ethics statement

Animal experiments were performed under IAUCUC protocol 2017A00000100 from The Ohio State University and protocol #31768–01 from The Lundquist Institute.

### *C. albicans* strains and karyotypic determination by whole genome sequencing

529L was obtained from prior work performed in the Anderson lab [30]. Selective genotyping by double digest restriction site-associated DNA sequencing (ddRAD-Seq) identified the Chr7 trisomy in 529L from our lab. Two additional 529L isolates were obtained from two other labs in the United States. Whole-genome sequencing of more than 20 colonies from these two strain populations revealed the presence of the same Chr7 trisomy. The same single-colony sequencing approach was then extended to the originally isolated 529L population, kindly provided by Dr. Julian Naglik (King's College, London, UK), wherein the diploid (Chr7x2) 529L strain was isolated as a colony. Whole genome sequencing of the Chr7x3 and Chr7x2 isolates of 529L, as well as the diploid SC5314 strain, was performed on the Illumina Nextseq 2000 platform at the Applied Microbiology Services Lab (AMSL) of The Ohio State University to an average depth of 80x coverage. The Chr7x3 SC5314 strains (AAB and ABB) were kindly provided by Dr. Judith Berman (Tel Aviv University, Israel), along with their whole-genome sequences and Y-MAP visualizations of their genome-wide copy numbers [8].

### Strain construction

Two identical plasmids containing the *NRG1* locus from SC5314 or 529L were constructed using gap repair cloning. The *NRG1* coding sequence and flanking intergenic regions were PCR-amplified from each strain with oligonucleotides encoding 20 bp of homology to the *NRG1* locus and 20 bp of homology to the cloning vector pSFS2A, and the majority of the pSFS2A plasmid was amplified as a linear sequence with reverse complemented primers to those for *NRG1*. Residual plasmid was digested with DpnI and both PCR products were purified before being transformed into chemically competent *Escherichia coli* DH5α cells. Primers for both SC5314 and 529L *NRG1* plasmids were the same. All resulting plasmids were sequenced via long-amplicon sequencing and were identical to the SC5314 *C. albicans* reference genome sequence or the 529L genome sequence. For integration of the third *NRG1* copy at the native *NRG1* locus, the plasmids were linearized with DraIII-HF and transformed into their corresponding strain background (*i.e.,* SC5314 *NRG1* into SC5314). Correct integration was verified by PCR, and the *SAT1*-Flp cassette was excised by plating to ~100 colonies on solid YPM medium top-spread with either 10 or 20 µg mL nourseothricin. Small colonies (indicative of loss of *SAT1*) were patched to YPD with and without 200 µg ml−1 NAT to ensure loss of the plasmid cassette and retention of *NRG1*x3.

### *In vitro* growth and fatty-acid sensitivity assays

Six independent cultures for each of the five *C. albicans* strains were grown overnight in liquid YPD at 30°C, before being diluted 1:200 into fresh medium. Growth was then traced as optical density under the same conditions in a 96-well plate, with absorbance readings every 15 minutes up to a total of 31 hours. Doubling time ('t_gen') and carrying capacity ('k') were determined via polynomial fits for the growth curves using the GrowthcurveR package in R [46]. Cultures to assay growth in the presence of undecanoic acid and undec-10-noic acid were prepared similarly with the exception of being diluted back 1:200 into YPD with the indicated concentration of medium-chain fatty acid dissolved in methanol at a final volume of 150 µL in 96-well plates. Three biological replicates were performed per concentration for each strain. Growth in the presence of fatty acids was assayed for 48 h, with absorbance readings obtained at 24 h and 48 h.

### *In vitro* filamentation assays

To assess filamentation in liquid media, three independent cultures per strain were grown overnight in liquid YPD at 30°C. Cultures were spun down, washed twice with 1x PBS, and diluted 1:100 in liquid RPMI in a final volume of 1 mL. After 3h of growth at 30°C, a 10-µL sample of the culture was imaged at 40x magnification using a light microscope. At least five independent field of view with a minimum of 50 cells per replicate was used to calculate filamentation frequency. Cells were manually counted and scored as yeast (round or oval cells) or filamentous (with non-budding projections).

Filamentation on solid media was assayed following the methodology of Dunn, *et al* [30]. Briefly, six biological replicates per strain were used for each media type (YPD or Spider agar medium) and temperature (30°C or 37°C) combination. Actively growing cells were counted and plated onto agar plates at a density of 80–120 cells per plate. Following 5 days of growth, the plates were imaged and scored using MIPAR image analysis software (MIPAR, Worthington, OH). Filamentation score for each replicate was calculated as the average ratio of radial filamentation area to the central colony area [(Area $_{total}$ – Area $_{central}$)/ Area $_{central}$] for all colonies in the plate.

## Interactions with human epithelial cell lines

Adherence and internalization of fungal cells into oral epithelial cells were determined as previously described [47]. Epithelial cells were grown to 95% confluency on fibronectin-coated glass coverslips in a 24-well tissue culture plate. Each coverslip was infected with $2 \times 10^5$ organisms in serum-free keratinocyte growth medium. After 2.5 h of incubation, the medium was aspirated, non-adherent organisms were removed by rinsing the coverslips with PBS, and cells were fixed with 4% paraformaldehyde. The adherent and non-internalized portions of the organisms were stained with rabbit anti-*C. albicans* antiserum (Biodesign International, Saco, ME) conjugated with Alexa 594 (Molecular Probes, Eugene, OR). Next, the host cells were permeabilized with 0.05% Triton X-100 for 10 min, and then all *C. albicans* cells were stained with rabbit anti-*C. albicans* antiserum conjugated with Alexa 488 (Molecular Probes, Eugene, OR). The coverslips were mounted inverted on a microscope slide and organisms were viewed under epifluorescence. Organisms were considered internalized if partially inside the epithelial cell. At least 100 organisms were counted per coverslip, and each experiment was repeated in biological triplicate, at least three times. Results were expressed as the number of internalized and cell-associated organisms per high powered field.

The human oral epithelial cell line OKF6/TERT-2 was kindly provided by J. Rheinwald (Harvard University, Cambridge, MA) and was cultured as previously described [48]. The extent of epithelial cell damage caused by the different strains of *C. albicans* was measured using the $^{51}$Cr release assay as previously described [49]. Briefly, oral epithelial cells were grown to 95% confluence in 96-well tissue culture plates and loaded with 5 µCi/ml $Na_2CrO_4$ (PerkinElmer) overnight. The unincorporated $^{51}$Cr was removed by rinsing with PBS, and then the epithelial cells were infected with $2 \times 10^5$ organisms suspended in serum-free keratinocyte growth medium. The organisms were incubated with the epithelial cells for 7 h. After the incubation period, the amount of $^{51}$Cr released into the medium and retained by the cells was determined by gamma counting. Each experiment was performed in biological triplicates with technical triplicates.

## Murine model of oropharyngeal candidiasis

The pathogenicity of the *C. albicans* strains were tested in the mouse model of oropharyngeal candidiasis as previously described with some modification [50]. Male C57BL/6 mice were obtained from Jackson Laboratory. For inoculations, immunocompetent mice were sedated with ketamine and xylazine, and a swab saturated with $2 \times 10^7$ *C. albicans* cells was placed sublingually for 75 min. After 1 and 5 days of infection, mice were euthanized and their tongues were harvested. The tongues were weighed, homogenized, and quantitatively cultured. To determine cytokine and chemokine protein concentrations during OPC the homogenates were prepared as previously described [51], and measured using a meso scale discovery multiplex ELISA array (Svar Life Science, Malmö, Sweden).

## Murine model of systemic candidiasis

Male and female C57BL/6 mice from the Jackson Laboratory were weighed and injected with *C. albicans* cells via tail-vein injections. For each strain, ten mice at 6 weeks of age (5 males + 5 females) were weighed and injected with $5 \times 10^5$ cells suspended in sterile saline solution to assess the mortality due to systemic infection. The mice were weighed daily and visually monitored twice daily until the end of experiment (14 days). Following the IUCAC protocol, mice were euthanized if they exhibited symptoms of severe illness and/or a decrease of ≥ 20% from the initial weight.

To assess the load of infection in internal organs, another set of ten C57BL/6 mice (5 males + 5 females) of similar age were injected similarly with $5 \times 10^5$ *C. albicans* cells. Following daily monitoring for two days, all mice were euthanized, and their brain, spleen, and kidney harvested. After weighing them, the harvested organs were mechanically homogenized and plated onto YPD plates by serial dilution to calculate colony forming units (CFU). The plates were incubated at 30°C for 48 h, subsequently imaged, and colonies counted to obtain the CFU load per gram of tissue.

### Statistical analyses and data visualization

All data were analyzed in R statistical computing platform v4.3.1 [52]. To assess the effect of aneuploidy and homolog identity on phenotypes, the data from a given assay were analyzed using separate ANOVAs (*aov* R package) for the three SC5314 strains and the two 529L strains. The exceptions were the liquid filamentation assay, where binomial generalized linear models (GLMs) were used to assess the proportional scale data (*glm* R package), and the systemic mortality assay, where the Log-Rank test was used to analyze the Kaplan-Meier curves (*survival* R package). In all cases, a statistically significant main effect was subjected to post-hoc analysis using the *emmeans* R package, which elucidated the pairwise differences. The graphical plots were all made using the *ggplot2* R package (v3.4.4).

### Supporting information

**S1 Fig. Secondary verification of the Chr7 trisomy in the two SC5314 aneuploid strains.** DNA was harvested from the SC5314 diploid and two aneuploid strains previously characterized as trisomic for Chr7 (8). A multiplex PCR assay was performed as in Arbour *et al.* [53], to verify that the received Chr7x3 AAB and ABB strains retained their Chr7 trisomy. Comparison of the amplicon abundances from Chr7x3 AAB and ABB strains against the known diploid SC5314 strain shows a distinct increase in DNA abundance for Chr7 (pointed black arrow).
(EPS)

**S2 Fig. Chr7 trisomy does not alter colony size on Spider medium.** The diameter of 20 colonies from Spider solid medium plates incubated at 37°C was measured for five different biological replicates used to measure filamentation.
(EPS)

**S3 Fig. Immune factor abundance from mouse tongue homogenates after oral *C. albicans* infection.** Concentrations of individual cytokines were determined from Day 1 tongue homogenates by a multiplexed ELISA assay across five assayed strains. * for $p < 0.05$, ** for $p < 0.01$, and *** for $p < 0.001$ based on one-way ANOVA followed by Tukey's HSD test.
(EPS)

**S1 File. All data combined into a single Excel file.**
(XLSX)

### Acknowledgments

The authors would like to thank the Anderson and Rappleye labs for useful discussion in the development of this manuscript. We would also like to thank Judy Berman for the kind gift of the Chr7 trisomic SC5314 strains and Julian Naglik for the kind gift of the original 529L population.

### Author contributions

**Conceptualization:** Abhishek Mishra, Scott G. Filler, Matthew Z Anderson.

**Data curation:** Abhishek Mishra, Norma V. Solis.

**Formal analysis:** Abhishek Mishra, Norma V. Solis.

**Funding acquisition:** Abhishek Mishra, Scott G. Filler, Matthew Z Anderson.

**Investigation:** Abhishek Mishra, Norma V. Solis, Siobhan M. Dietz, Audra L. Crouch, Matthew Z Anderson.

**Methodology:** Abhishek Mishra, Norma V. Solis, Siobhan M. Dietz, Scott G. Filler, Matthew Z Anderson.

**Project administration:** Scott G. Filler, Matthew Z Anderson.

**Resources:** Scott G. Filler, Matthew Z Anderson.

**Software:** Abhishek Mishra.

**Supervision:** Scott G. Filler, Matthew Z Anderson.

**Validation:** Abhishek Mishra, Norma V. Solis.

**Visualization:** Abhishek Mishra, Norma V. Solis.

**Writing – original draft:** Abhishek Mishra, Matthew Z Anderson.

**Writing – review & editing:** Abhishek Mishra, Siobhan M. Dietz, Scott G. Filler, Matthew Z Anderson.

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
