## [Decision Letter · Decision Letter 0]

Dear Matt,

We are pleased to inform you that your manuscript entitled "Strain background interacts with chromosome 7 aneuploidy to determine commensal and virulence phenotypes in Candida albicans" has been editorially accepted for publication in PLOS Genetics. Congratulations!

Yours sincerely,

Anja Forche, PhD

Guest Editor

PLOS Genetics

Eva Stukenbrock

Section Editor

PLOS Genetics

Aimée Dudley

Editor-in-Chief

PLOS Genetics

Anne Goriely

Editor-in-Chief

PLOS Genetics

Comments from the reviewers (if applicable):

The manuscript submitted to PlosGenetics had undergone substantial revisions after a previous submission to a different journal. The authors did a very thorough job to address all concerns/critics and therefore the version submitted to PlosGenetics was reviewed very positively by both reviewers and the editor.

Reviewer's Responses to Questions

**Comments to the Authors:**

Reviewer #1: Mishra et al. show that the phenotypic effects of aneuploidy in C. albicans depend on the strain background. The revised version addresses the points by previous reviewers sufficiently; while it could be argued, that the manuscript is largely descriptive and no clear mechanism explaining the differences between the strains was identified, the authors did investigate one possible mechanism (NRG1 copy number) in detail. I consider this to be sufficient as it is possible that not a single mechanism but the cumulative effect of a multitude of SNPs/allelic differences drive the phenotypic differences.

Reviewer #2: This manuscript has already undergone extensive review at another journal. As a result, it is complete, the experiments are well conducted, and the data presentation is clear.

The study examines the effects of Chr7 trisomy on Candida albicans colonisation and virulence across two different strain backgrounds, SC5314 and 529L. The authors show that strain background and homolog identity can influence phenotypic outcomes. However, the phenotypes observed are generally weak. While previous reviewers have commented that the work remains largely descriptive, and that the mechanistic underpinnings are only superficially addressed, it is my opinion that this manuscript makes a valuable contribution to the field. It adds to our understanding of C. albicans strain diversity and the broader role of copy number variation. The exploration of NRG1’s involvement, although mostly negative, is also interesting.

Overall, I support publication. The article is complete and, despite its descriptive nature, will be of interest to researchers working on C. albicans.

**Have all data underlying the figures and results presented in the manuscript been provided?**

Reviewer #1: Yes

Reviewer #2: Yes

PLOS authors have the option to publish the peer review history of their article (what does this mean? ). If published, this will include your full peer review and any attached files.

**Do you want your identity to be public for this peer review?** For information about this choice, including consent withdrawal, please see our Privacy Policy .

Reviewer #1: No

Reviewer #2: No

**Data Deposition**

http://datadryad.org/submit?journalID=pgenetics&manu=PGENETICS-D-25-00272

**Press Queries**

---

## [Editor Report · Acceptance letter]

PGENETICS-D-25-00272

Strain background interacts with chromosome 7 aneuploidy to determine commensal and virulence phenotypes in Candida albicans

Dear Dr Anderson,

We are pleased to inform you that your manuscript entitled "Strain background interacts with chromosome 7 aneuploidy to determine commensal and virulence phenotypes in Candida albicans" has been formally accepted for publication in PLOS Genetics! Your manuscript is now with our production department and you will be notified of the publication date in due course.

With kind regards,

Anita Estes

PLOS Genetics

On behalf of:
